# Effects of Post Anthesis Foliar Application of Sodium Selenite to Soybeans (*Glycine max*): Lipid Composition and Oil Stability

**DOI:** 10.3390/biom9120772

**Published:** 2019-11-23

**Authors:** María José Escalante-Valdez, Daniela Guardado-Félix, Sergio O. Serna-Saldívar, Daniel Barrera-Arellano, Cristina Chuck-Hernández

**Affiliations:** 1Tecnologico de Monterrey, School of Engineering and Sciences, Eugenio Garza Sada 2501, Col. Tecnologico, C.P. 64849, Monterrey, N.L., Mexico; majoescalant@gmail.com (M.J.E.-V.); daniela.guardado@tec.mx (D.G.-F.); sserna@tec.mx (S.O.S.-S.); 2Programa Regional de Posgrado en Biotecnología, Facultad de Ciencias Químico-Biológicas, Universidad Autónoma de Sinaloa, FCQB-UAS, AP 1354, C.P. 80000 Culiacan, Sinaloa, Mexico; 3Department of Food Technology, Faculty of Food Engineering, University of Campinas (UNICAMP), Cidade Universitária Zeferino Vaz, P.O. Box 6121 Campinas, Brazil; danbramex@gmail.com

**Keywords:** selenization, phytosterols, tocopherols

## Abstract

This study aimed at determining whether applying selenium to soybean plants affected composition and oil oxidative stability of the seeds. Soybean was cultivated and sodium selenite (Selenite) added by foliar application (0, 200, or 300 g Selenite/Ha). Physical and chemical characterization was performed on the harvested seeds (thousand seed weight, bulk and true densities, fat, fiber, ash, protein, nitrogen free extract and selenium content). Soybean oil was tested in terms of Oxidation Induction Time (OIT), fatty acid, tocopherols, phytosterols, density, refractive index and saponification and iodine values. All seeds showed similar composition: crude fat (around 20%) and crude fiber (from 8.4 to 9.3%). Control seeds and those treated with 200 g Selenite/Ha contained higher protein concentration (37%), compared to the 300 g treatment (35.9%). All seeds showed similar ash content (7%). OIT values for both treatments were slightly lower (from 39.1 to 43.7 min) compared with 45.02 min in the control. Polyunsaturated fatty acids were higher for the 300 g Se/Ha (50.2%) compared with 48.2 to 49.4%of the other treatments. All samples showed similar phytosterols and tocopherols concentrations. Results showed that OIT values maintained an inverse relationship with selenium content, suggesting that foliar fertilization enhanced oil oxidation or acted as a pro-oxidant at the applied rates.

## 1. Introduction

Soybeans (*Glycine max* L) is the most important legume–oilseed crop worldwide with a yearly production that exceeds 352 million tons planted on more than 123 million hectares [1]. Seeds are commonly processed to extract the highly polyunsaturated crude oil and a high protein quality defatted meal. The seed contains from 17 to 21% oil abundant in linoleic acid and 35 to 36% protein rich in globulins. Generally, soybeans are soil-fertilized with nitrogen, phosphorus, and potassium in order to increase yields and, in some instances, crops are enhanced by foliar fertilization in order to provide essential micronutrients that further increase yield, and can modify the mineral and nutritional profiles of harvested grains. One typical example is the soil or foliar application of selenium (Se).

Recently, Se, both in its inorganic and metabolized organic forms, has attracted special attention because of its proven positive role in nutrition and health. The recommended daily allowance of this essential trace mineral is of only 70 and 55 µg for adult men and women, respectively. Se is essential because it is required to synthesize selenoproteins and enzymes for important metabolic functions. Its deficiency leads to a higher risk of cancer and impaired immune function, and it has also been associated with Keshan and Kaschin–Beck diseases and cardiomyopathy. All these ailments are strongly associated with diminished antioxidant protection [2]. Se is provided through the diet in both organic and inorganic forms, the first being the most bioactive and less toxic. Selenomethionine (SeMet) and selenocysteine (SeCys) are considered the main organic moieties. These amino acids are homologous of methionine (Met) and cysteine (Cys), respectively. Among selenium-containing amino acids, SeMet, present in grains, is considered the major form (>65%) [3]. SeCys is the organic form most commonly found in animal tissues and is synthesized from SeMet or selenite [4]. It is known that SeCys has better accessibility to the oxidation stage compared to Cys and therefore possesses better redox capacity. This is the reason proteins containing SeCys instead of Cys possess active sites with 100 times better catalytic efficiency. The most relevant selenoproteins are enzymes Glutathione Peroxidase (GPx), deiodinases (DIO), and thioredoxin reductases (TR). GPx are composed of a group of five antioxidant enzymes, which catalyze reduction of hydroperoxides, whereas DIO play a key role in the activation or deactivation of the thyroxine through the reductive deiodination. TRs reduce selenites and selenates to generate precursors for selenoproteins in order to keep the pool of thioredoxin stable [5]. In short, these enzymes protect mammalian systems from oxidative stress. It is well-recognized that Se forms part of several key proteins that contain SeCys residues in their structures [6,7]. Furthermore, supra-nutritional intakes have long been linked to the prevention of cancer [8] and more recently in the prevention of cardiovascular diseases [9].

Plants can effectively store Se in their grains and fruits, mainly in the form of SeMet. Levels vary greatly depending on the growth region. The main agronomic factor affecting Se concentration in seeds is the amount of this mineral present in the soil. Bio-fortification refers to either the addition of selenium as part of the nutrients purposely added to plants or to the plant breeding or genetic modification of crops. The easiest and simpler way to apply Se is through soil or foliar applications using inorganic sources such as Na_2_SeO_4_ and/or K_2_SeO_4_. These two inorganic salts effectively and rapidly increase Se concentration in plant tissues [10]. This strategy has been successfully used in wheat [10], barley [11], green tea [12], and olives [13]. In olives, an oil-bearing fruit, the concentration of selenium increased about 50 times and helped to significantly improve yields, especially when trees underwent water stress. More importantly, authors also observed that the olive oil had better oxidative stability, likely due to the higher concentration of natural phenolic compounds. In addition, the concentrations of carotenes and chlorophyll also increased. Finally, authors concluded that Se did not affect the characteristics of the olive fruit nor the organoleptic properties of the extra-virgin olive oil. Our research group [14] reported the improvement in the oxidative stability of chickpea oil after germination with selenium. The oil content of this pulse is around 6% of its weight. Regarding soybean, there is scarce information related to its biofortification using selenium. Soil or foliar application of sodium selenite to two soybean varieties (Wandou No.2 and Tong’ai No.405) using two concentrations (200 or 300 g Selenite/Ha) has been tested [15]. The main findings of their study indicated that the application of Se did not significantly affect yield nor protein and oil concentrations except when this trace mineral was sprayed on the leaves. The foliar application increased the concentration of seed lipids and was more efficient compared to soil fertilization in terms of Se content in the seeds. The difference is attributed to the better transfer and accumulation when applied through the leaves. The soil application is less efficient because Se must be first absorbed and then translocated to other plant tissues. Another important factor to consider is the soil pH because it is known that acidic soils enhance the binding of sodium selenite to insoluble compounds reducing its bioavailability.

The application of selenium as reported by [13,14,15], increase its concentration on seeds ensuring daily dietary selenium intake, preventing diseases related to its deficiency and enhancing oil oxidative stability and lipid content. These results encouraged us to conceive this study using the most relevant oilseed annual crop of the globe. Thus, this investigation was undertaken to assess the effects of sodium selenite applied through foliar fertilization to soybean after anthesis in terms of physical and chemical composition of seeds, Se concentration, oxidative induction time of extracted oils, and concentration of relevant phytochemicals, which included phytosterols and tocopherols.

## 2. Materials and Methods

### 2.1. Agronomic Experimental Design and Selenium Application

Non-genetically modified (non-GMO) soybean, variety Vernal was planted in the agricultural station of Tecnologico de Monterrey located in Hualahuises, Nuevo León, México (24°56′N 99°42′W). Twelve different plots of 6.5 m × 6.5 m (42.25 m^2^) randomly distributed in 4 blocks were prepared. Each block contained three plots that were randomly assigned to the control or experimental treatments consisting of the application of 0, 200 or 300 g of Selenite/Ha. The doses were selected from a previous study in soybeans [15]. Soil fertilization was not applied and weeds were manually controlled. Two foliar applications of sodium selenite (Na_2_SeO_3_) were made post-anthesis 10 and 11 weeks after sowing. In order to avoid the plot edge effect, all plants growing in the periphery of the plots were discarded. Mature seeds from usable plots were harvested 23 weeks after sowing and then manually cleaned to remove foreign material, splits, and immature and damaged seeds. The cleaned seeds were placed in sealed plastic bags and stored at −18 °C until analyses.

### 2.2. Height of Soybean Plants and Soybean Yield

The plant height was measured weekly from the base of the plant to the apex using a measuring tape. Yield was obtained by extrapolating the harvested seeds of each plot to hectare and expressed as kg/Ha.

### 2.3. Physical Characterization of Soybeans

The bulk density of seeds, also known as test weight, was determined with the Winchester Bushel Meter, using the 500 mL cup, according to method 55-10 of the American Association of Cereal Chemists [16]. Briefly, the cleaned soybean seeds were placed in the hopper or Cox funnel of the Winchester Bushel Meter with enough seeds to fill the cup, making sure the gate was closed and aligned right in the center of the empty cup. Then, the gate was opened in order to allow seeds to flow and overfill the cup. Afterward, excess grain over the cup´s rim was removed with the aid of a strike off stick by a zigzag motion and the seed weighed with an accuracy of 0.1 g. The test weight was expressed in kg/cubic meter (kg/m^3^). True density was determined with the multipycnometer (Quantachrome Instruments, Boynton Beach, FL, USA) by helium displacement according to the method described by [17].

The 1000 seed weight was estimated by randomly selecting 100 cleaned seeds that were weighed with an accuracy of 0.01 g. The resulting weight was multiplied by 10 as reported by [18].

### 2.4. Chemical Analyses of Soybeans

Moisture, crude fat, crude fiber, crude protein, ash, and nitrogen free extract (NFE) were determined in whole seed samples by triplicate. Moisture was gravimetrically assayed according to method 925.10 of the [19] whereas crude fat with the Goldfisch gravimetric method 30-20.01 of the [16]. Protein and ash contents were assayed following the micro Kjeldahl procedure 978.02 and gravimetric incineration (550 ± 3 °C) method 923.03 of the [19], respectively. NFE values, which gave an approximation of non-fibrous carbohydrates, were obtained by subtracting moisture, crude fat, crude fiber, crude protein, and ash from 100. 

Selenium concentration was assayed from 500 mg samples digested with 10 mL 70% nitric acid in a microwave oven (Mars 5 CEM, Matthews, NC, USA) for 20 min at 200 °C according to the method previously reported by [20]. The resulting hydrolysates were brought to 20 mL with HPLC grade water. Selenium concentration was determined with Inductively Coupled Plasma Mass Spectrometry (ICP-MS, X Series2, Thermo Scientific, Waltham, MA, USA) equipped with a concentric glass nebulizer type C (Meinhard, Golden, CO, USA). The internal standards were 165Ho and 159Tb and the reaction gas was He with 7% hydrogen. Results were expressed as µg Selenium/g (dry weight).

### 2.5. Oil Extraction

Before oil extraction, whole seeds belonging to the different treatments were ground into a coarse meal in a coffee mill (Krups GX4100, Mexico City, Mexico). The oil was extracted from each of the whole meals in an incubator (RF 1575, VWR International, Cornelius, OR, USA) using hexane at the ratio of 1:3 at 40 °C and 100 rpm of agitation. The extraction ran for 5 days and the solvent was replaced every 12 h. At the end of the extraction, the crude oils and hexane were recovered in a rotary evaporator (R-220, Büchi, Flawil, Switerlad) equipped with a water bath operating at 60 °C and 300 mbar. In order to prevent oxidation, the crude oils were then placed in dark bottles, flushed with nitrogen (N2-14A-K727, 5183-2004 Nitrogen Generator, Agilent, Palo Alto, CA, USA) and stored at −20 °C.

### 2.6. Physicochemical Characterization of Soybean Oil

Oil density was determined by weighing the oil at 20 °C, in a 25 mL volumetric flask. Values were calculated by dividing the oil weight (g) by the volume (mL). Oil refraction index (RI) was determined in a digital refractometer (PAL-1; Atago, Tokyo, Japan) at a temperature of 23 °C [21].

The iodine index, which is related to the number of unsaturation or double bonds in fatty acids, was determined according to [21,22]. Values were calculated after determining the fatty acid profile of each oil and multiplying each fatty acid by its constant.

The saponification value, which is related to the length of the fatty acids that make up triglycerides, was also assessed by multiplying the fractional amount of each fatty acid by their molecular weight and the result divided by 100.

### 2.7. Oxidation Induction Time (OIT) 

OIT values were determined using the differential scanning calorimeter (DSC) according to [23]. A thermal analyzer (TA Q2000, New Castle, DE, USA) coupled to a refrigeration system RCS90 (TA Instruments, Waters LLC, New Castle, DE, USA) was used. Assays were carried out using 2–4 mg of each of the oils extracted from the control and experimental seeds, weighted in open aluminum pans. The temperature program used was sample equilibrium to 60 °C, holding this temperature for 5 min, ramping at a rate of 20 °C/min to a maximum temperature of 200 °C, which was maintained for 5 additional minutes. To create an oxidative atmosphere, air–flow rate was set at 50 mL/min.

### 2.8. Fatty Acids Composition

The fatty acid composition of the control and experimental soybean oils were determined in a gas chromatograph (GC) following the official method Ca 12-55 of the [24]. The equipment (Agilent series 6850 GC System, Santa Clara, CA, USA) was equipped with a capillary column (Agilent DB-23, 60 m length, internal diameter 0.25 mm and 0.25 µm thickness) and a flame ionization detector (FID). Helium was used as carrier at a flowrate of 1.0 mL/min and linear velocity of 24 cm/s. Samples (1.0 µL) were injected into the port, with a temperature of 250 °C, whereas the temperature of the detector was set at 280 °C. Analyses were performed at an oven temperature program of 110 °C during 5 min followed by ramping up to 215 °C at a rate of 5 °C/min and maintaining this maximum temperature for 24 min. 

### 2.9. Tocopherols and Phytosterols in Soybean Oils

For tocopherol analysis, an HPLC system (PerkinElmer 200 Series, Norwalk, CT, USA) with a fluorescence detector (PerkinElmer 200a Series, Norwalk, CT, USA) was used. The detector was set to excitation and emission wavelengths of 290 and 330 nm, respectively. The column was a Hibar RT 250 × 4 mm (Li Chrosorb Si 60, 5 µm). Hexane: isopropanol (99:1) was used as the mobile phase at a rate of 1.0 mL/min.

The phytosterol composition of soybean oils was determined with a gas chromatograph (GC, Agilent 68650 Series GC System, Santa Clara, CA, USA) equipped with a capillary column Rxi-5HT (diphenyl dimethyl polysiloxane) of 30 m long, 0.25 mm internal diameter, and 0.25 µm thickness. Helium was used as carrier gas at a flow rate of 1.0 mL/min and a linear velocity of 33 cm/s. Samples (2.0 µL) were injected into the injection port set at a temperature of 300 °C, and the temperature of the flame ionization detector was also fixed at 300 °C. Analyses were performed at an oven temperature of 265 °C for 40 min. 

### 2.10. Statistical Analysis

All measurements were made in triplicate and results were expressed as averages ± standard deviations. The statistical analyses were made with ANOVA (analysis of variance) procedures and differences of means were assessed using Tukey tests (α = 0.05, Minitab 17, Minitab Inc., Coventry, UK).

## 3. Results

### 3.1. Agronomic Features of Soybean and Characterization during Seed Development

#### 3.1.1. Height of Plants at Various Weeks after Sowing

Figure 1 shows soybean plant height. Plants were 60 cm at week 8 after sowing, growing to 120 cm at week 12. All the plants, control and treatments 200 or 300 g Selenite/Ha, were of similar heights. All plants grew about 20 cm weekly from weeks 8 to 10. After the sodium selenite application in week 10, plant’s growth slowed down for one week (week 11), but this behavior cannot be associated with the selenium foliar application since the control plants grew at the same rate. The second foliar application was made at week 11. All plants recovered the growing rate of 20 cm weekly at week 12.

#### 3.1.2. Density, Thousand Seed Weight, and Yield per Hectare

Density of seeds is an agronomic measurement useful to assess the quality of grain. Soybean seeds showed a bulk density ranging between 693.3 and 700 kg/m^3^ (Table 1), similar to that reported in the literature [18,25]. At the same time, real density had a range of 1240.0 to 1242.1 kg/m^3^ for all treatments, similar to reported values [25,26,27]. Table 1 shows the thousand seed weights; the values were slightly lower than literature reports [28,29] with weights ranging from 115 to 122 g, with no statistically significant differences among treatments.

Seed yield, calculated as the ratio of seed weight over sowed area, is a decisive factor for agronomical development. In this study, the production yield was not affected after selenium application, and the control and experimental treatments yielded similar amounts of seed, which ranged from 1179.8 up to 1268.0 kg/Ha (Table 1). When the production of oil was calculated based in the seed yield per hectare and the crude oil content for each type of seed, results indicated a slightly higher amount of oil for both selenium treatments, compared with the control: 245.4, 279.0 and 267.2 kg oil /Ha for 0, 200 and 300 g/Ha treatments, respectively.

#### 3.1.3. Chemical Characterization of Selenized Soybeans

The composition of the soybean whole meals as well as selenium concentration are shown in Table 2. The main component of soybean meal was protein: 37.6% for control seeds and 37.5% for the 200 g/Ha treatment, whereas, for the 300 g/Ha flour, a lower protein percentage (35.9%) was observed. The oil fraction can be considered the second most important in economic terms, yielding similar amounts (20.8, 22.0, and 22.1% for control, 200 and 300 g Selenite/Ha, respectively). Crude fiber (9.3, 9.0, and 8.4%), ash (7.1, 7.2, and 7.2%) and free nitrogen extract (25.2, 24.3, and 26.4%) contents were also similar for seeds harvested from the 0, 200 and 300 g/Ha plots, respectively. 

As expected, the contents of selenium in the seed varied according to the amount of foliar fertilization (Table 2). The concentration of this mineral varies according to its content in soil, geography, and type of plant food. The soybean seeds analyzed contained 1.0, 18.9, and 23.35 µg of Se/g seed fertilized with 0, 200, and 300 g/Ha, respectively. All treatments showed statistical differences indicating a positive association between the applied sodium selenite and its accumulation in the seed. The efficiency of absorption (selenium taken by the seeds versus selenium applied) for 200 and 300 g/Ha was 6.16 and 4.97%, respectively.

In order to evaluate the effect of foliar application of selenium in soybeans, the oil characteristics and oxidative stability were assessed, and results are shown in the following sections.

### 3.2. Soybean Oil General Characterization

Soybean oil is considered the most used vegetable oil worldwide. Despite its importance, data about its physical properties are scarce and were mainly reported in older studies [30]. 

Table 3 summarizes important data related to soybean oil: density, refraction index (RI), iodine, and saponification indexes. The density of oils at 20 °C of the control and selenized seeds varied in a range from 0.9029 to 0.9097 g/mL, with no significant differences among treatments, indicating that foliar Selenite fertilization did not affect oil density values. These results were similar to those reported by [21,30] at 40 °C. 

Results of the RI (Table 3) indicated similar values for all the oils, varying from 1.4685 to 1.4718. The results obtained in the control and 300 g Se/Ha treatments agree with those reported in other studies where soybean oil was evaluated at room temperature [21,30], whereas the 200 g Selenite/Ha treatment presented a slightly lower RI. Likewise, the iodine value that is related to the degree of unsaturation and susceptibility to lipid peroxidation [21,22] indicated no significant differences among treatments with values ranging from 118.07 to 119.73 (Table 3). On the other hand, saponification values, which are related to the molecular weight of triglycerides and defined as the amount in milligrams of potassium hydroxide (KOH) required to saponify 1 g of oil [22], were in the range of 192.42 to 193.39 mg of KOH.

### 3.3. Oxidation Induction Time (OIT)

The determination of OIT is considered vital in the oil industry because this index predicts the oxidative shelf lives at different temperatures. The factors that affect OIT are geometry, distribution, and presence of double bonds. Vegetable oils, particularly those rich in polyunsaturated fatty acids, are recognized by their positive health benefits especially in terms of reducing the risk of cardiovascular diseases [31]. However, the main disadvantages are that they are more prone to oxidation because of the higher degree of unsaturation or iodine value, and more susceptible to yield peroxides, which produce undesired flavors and odors [21].

When assessing the effect of selenization on olives [32] and chickpea sprouts [33], an improved oxidation stability was observed. These results encouraged us to evaluate if soybean oil OIT values can be increased through the stress induced by the application of selenium, so that a higher shelf life can be achieved without the use of natural or synthetic antioxidants, such as butylated hydroxyanisole (BHA) or butylated hydroxytoluene (BHT). These synthetic antioxidants have been frequently questioned regarding their toxicity and are considered non-GRAS [31].

OIT results are presented in Figure 2 for all oils extracted from soybeans (control, 200 and 300 g Selenite/Ha). The three points in the graph correspond to the inflection time in minutes. Oils extracted from control seeds had the highest OIT with 45.02 min followed by the counterpart from the 200 g Selenite/Ha (43.67 min) treatment. Interestingly, oils from the 300 g Se/Ha presented the lowest stability with 39.09 min. Unexpectedly, an inverse association was observed between the use of selenium in the different samples with OIT; we found that the higher the amount of selenium fertilization, the lower the OIT in soybean oils. OIT results are shown in Figure 2.

In order to understand the observed OIT differences, the fatty acid composition of the oils was assessed, and results are described in the following section.

### 3.4. Fatty Acid Profiles

The fatty acid profiles of the different oils are shown in Table 4. The control and selenized oils showed similar average fatty acid compositions with approximately 17% saturated fatty acids, whereas monounsaturated represented 34.3%, 32.7%, and 31.8% for the treatments with 0, 200, and 300 g Selenite/Ha, respectively. As expected, the polyunsaturated fatty acids were the highest fraction in the control and experimental oils. The amounts of polyunsaturated fatty acids increased as the selenium fertilization incremented, so oils obtained from the treatment with 300 g Se/Ha contained approximately 2% more compared to the control.

Palmitic (C16:0) and stearic (C18:0) were the most abundant saturated fatty acids, whereas oleic (C18:1, ω9) is the most predominant monounsaturated. The polyunsaturated linoleic (18:2 Δ 9,12 ω6) was the most abundant fatty acid with values exceeding 40% of the total composition. The omega-3 linolenic (18:3 Δ 9,12,15 ω3) constituted more than 5% of the total fatty acid composition (Table 4). Needless to say, these two polyunsaturated fatty acids are the most prone to oxidation.

### 3.5. Tocopherol and Phytosterols Content in Soybean Oil

Tocopherols are potent antioxidants in vegetable oils at low concentrations and protect polyunsaturated fatty acids from lipid peroxidation. In addition, they protect humans from oxidative stress and cancer [34]. In soybean oil, tocopherols can be found as α, β, γ, and δ, the last two with the highest antioxidant efficiency [30]. Figure 3 shows tocopherol contents. γ-tocopherol was slightly higher, with 69.6% of the total content, in the 300g Selenite/Ha treatment, while it represented 69.0 and 68.2% of the control and 200 g Selenite/Ha counterparts, respectively. The second most abundant was the α-tocopherol. The treatment with 200 g Selenite/Ha contained the highest amount with 16.7% followed closely by the 300 g Selenite/Ha (16.15%) and control (15.1%) oils. The third most abundant tocopherol was the δ with 14.3% for control followed by oils extracted from the 200 and 300 g Selenite/Ha treatments, which contained 12.9 and 12.8%, respectively.

Finally, β tocopherol had the lowest concentration, with 1.52, 2.13, and 1.44 % of total tocopherols associated with the control, 200 and 300 g Selenite/Ha oils, respectively.

Sterols are minor components in vegetable oils, but they are considered the most important non-saponifiable lipidic fraction [30] and have recently received attention due to their proven hypocholesterolemic effects [35,36]. Total phytosterols in soybean oils were in a range of 3343 to 4231 mg/kg. The lowest amount was found in oils from the 300 g Selenite/Ha, followed by control and 200 g Selenite/Ha. As described in Figure 4, the most abundant phytosterol was the *beta*-sitosterol + sitostanol, in a range from 48.6 to 56.1% of total phytosterols. The second most relevant was campesterol (16.5–22.7%), followed by stigmasterol (16.0–17.2%). Even when there were no differences among treatments, a small increase in *beta*–sitosterol + sitostanol was observed in the oil extracted from the 300 g/Ha treatment.

The same trend was observed in campesterol associated with the 200 g/Ha treatment and stigmasterol for the control oil. These results show that biofortification with selenium slightly affected the phytosterol composition of soybean oils.

## 4. Discussion

### 4.1. Agronomic Attributes of Soybean and Characterization of the Seed

#### 4.1.1. Height of Selenized Plants at Various Weeks after Sowing

Plant height is a feature related to plant growth that indicates the availability of nutrients as nitrogen and phosphorous [37,38]. The height is also influenced by other factors, such as vigor of the seeds and number of plants cultivated per area [38]. According to [39], this is a quantitative attribute controlled by multiple genes and affected by the environment. In this work, the height of the plants reached around 120 cm, higher than the range of 86.61 to 99.19 cm reported by [39]. The effect of phosphorous and seed size over the growth of the plant was tested by [38], obtaining heights up to 161.9 cm. The relatively high size of the soybean plants tested in our work was associated with the availability of resources in the field and good agronomic practices especially in terms of irrigation. 

Agronomic practices, such as the number of plants sown by area, are considered relevant factors influencing the morphology of the plant and the yield. Increasing plant density in a given area increases height and lodging, while reducing branching and the number of pods per plant, but allows more pods and beans per unit area. However, as lodging increases, yield decreases. The optimum plant population is different depending on varieties and environments, so, in this case, the higher size of the plants compared with previous reports could have been influenced by the plant density sown in each experimental plot.

#### 4.1.2. Density, Thousand Weight, and Yield per Hectare

Density results (693.3 to 700 kg/m^3^) were similar to values reported in the literature, whereas thousand seed weights (115 to 122 g) were slightly lower than previous reports. These attributes are intimately associated with the seed moisture content, hardness, insect and mold damage, as well as oil and protein contents. The higher the amount of oil, the lower the density and vice versa for protein concentration [17,18,28]. In the case of thousand seed weight, this is an indicator of the seed size, the relative amounts of the seed anatomical parts and an indicator of the production yield of flours and derivatives [40,41]. This characteristic is also associated with the soybean yield and germination capacity or vigor [38]. The lower thousand-seed weight is an attribute that greatly depends on the cultivar or genotype. Despite these results, there were no statistical differences among treatments, indicating that the foliar application of selenium did not affect this characteristic, nor the true and bulk seed densities.

Regarding yield, all the plants in the study yielded similar amounts of seeds, although the experimental treatments tended to produce slightly, but non-significant, higher amounts of seeds. This was likely linked to the antioxidant protection induced by Selenite. This trace mineral improves the resistance to different types of abiotic stresses through several mechanisms, as ascorbate production and glutathione peroxidase activity, detoxifying because of the effect over Reactive Oxygen Species (ROS). These mechanisms improve the crop resistance during growth [10]. 

In Mexico and the United States of America, the average yields of soybean per hectare in 2017 were 1648 and 3299 kg/Ha respectively [1], higher than results presented here. The differences could be due to the fact that most of the soybean crops planted in the United States of America are genetically modified or GMO, which commonly produce higher yields. The genotype used in the present study was a non-GMO cultivar (Vernal). As already described, the seeds used had a low thousand weight, indicating a smaller seed size compared with other alternatives. Seed size is very important in terms of productivity [42,43]. 

#### 4.1.3. Chemical Characterization of Selenized Soybean Seeds

The most important component in soybean seeds was protein (Table 2). According to the literature, the distinctive composition of soybean includes 40% protein followed by 20% fat, 25% carbohydrates, 10% moisture, and 5% ash. These components change slightly according to the cultivar [29,44] and environmental conditions prevalent during seed growth. Results shown in Table 2 are similar to values reported by [29] and in the USDA database [45] in terms of protein, fat, fiber, minerals, and NFE. The exception was the seeds harvested from plants fertilized with 300 g/Ha specifically in terms of protein concentration, which was significantly lower compared with the other treatments. These results suggest that the highest level of Selenite fertilization may have interfered with anabolism or protein synthesis during seed development after anthesis. 

As expected, the foliar fertilization with sodium selenite greatly changed the content of this mineral in seeds. The amount of soybean seeds to fulfill the Recommended Daily or Dietary Allowance (RDA) of 55 µg of selenium were 55 g, 2.91 g and 2.35 g for the control, 200 and 300 g/Ha, respectively. On the other hand, the Tolerable Upper Intake Levels (UL), estimated at 400 µg/day, were reached with 400 g, 21.6 g, and 17.09 g for the same treatments. These data clearly indicate that the proposed selenization strategy was effective but potentially toxic when daily intakes of seeds exceed 20 g. Therefore, it is recommended to investigate, in future studies, lower dosages of foliar application and determine if the Se is inorganic or incorporated into organic molecules such as SeMet or SeCys. It is well-known that organic forms of selenium are more bioactive and less toxic to mammals [46].

Values of selenium concentrations in soybean seeds harvested in the USA varied from 0.07 to 2.9 µg/g. The Se content of control seeds fell within that range (Table 2). Needless to say, the selenized seeds contained higher concentrations compared to control counterparts. The levels were even higher compared to biofortified seeds assayed by [15]. These authors found concentrations from 6.3 to 7.9 µg selenium per gram after fertilizing with sodium selenite via foliar application. The difference with the present study is attributed to the higher and double application of sodium selenite post-anthesis.

The efficiency of absorption of selenium applied by foliar treatment was, as previously described, 6.16 and 4.97% for seeds fertilized with 200 and 300 g Selenite/Ha. The foliar application was less efficient for selenization when compared with other methods, such as fermentation and germination, techniques that are known to incorporate about 80% of the applied selenium [20]. 

### 4.2. Soybean Oil General Characterization

Density is one of the most relevant physical characteristics of the oil. It decreases as the temperature rises and increases when the degree of unsaturation is higher. In addition, density maintains an inverse association with the molecular weight of triglycerides [30]. Average density for all treatments showed no difference compared with the control oil. 

RI is another physical parameter that is commonly evaluated in oils and is based on the property of refraction. It is useful for establishing oil purity and for identifying the progress of reactions such as catalytic hydrogenation [47]. RI is influenced by the molecular weight of the sample, unsaturation degree, conjugation, presence of solids, temperature, and fatty acids chain length [22,30]. In this study, the oil obtained from seeds fertilized with 200 g Selenite/Ha showed a lower RI compared to the other treatments, likely due to oil oxidation during storage. It was observed that the tocopherols content (the main fat-soluble compound in the cellular antioxidant defense system) of this treatment was slightly lower compared to the 0 and 300 g Selenite/Ha samples, suggesting that it was less protected against oxidation.

Iodine values were lower than the average reported typically for soybean oil, which averages 130% [30]. The relatively low iodine values would explain the observed lower density in the selenized soybean oils. Saponification values were calculated from the fatty acid composition and showed normal results, within the range reported for soybean oils (189 to 195 mg KOH/g) [21,30]. Similar saponification values indicate that triglycerides had similar molecular weights independent of the selenization.

### 4.3. Oxidation Induction Time (OIT)

Previous studies [13,32] analyzed extra virgin oil from olives selenized by foliar application, finding that the oxidation stability increased after fertilization during the olive’s development under drought stress. In these conditions, ROS were produced and selenium acted as antioxidant blocking free radicals. In the studies mentioned above, the authors associated the increase in oxidation stability to the higher content of phenolic compounds, which have been identified as the main antioxidants in crude oils. Oxidative stability of oils from selenized chickpea sprouts was evaluated [14], finding higher stability attributed to the decrease in the activity of lipoxygenases (LOX). LOX contributes to the growth, development, and resistance to pests and to mobilization of lipid stores. When the plant or the seed suffers a lesion, LOX catalyzes the deoxygenation of polyunsaturated fatty acids producing conjugated mono hydroxy peroxides. In selenized chickpea sprouts at different concentrations (1 and 2 mg Na_2_SeO_3_/100 g seeds), LOX activity was reduced from 55 to 67%, respectively, compared to the control, possibly because, when the hydroxy peroxides were reduced in the presence of selenium, the enzyme activation did not occur. The increase of the oxidative stability was also attributed to the increment of carotenoids and the production of phenolic compounds in response to stress, demonstrated by the increment in the cellular antioxidant capacity [20]. It is worth noting that, in both processes of selenization (foliar application in olives and germination in chickpea), the stress induced was two-fold, combining the stress by selenium with drought or germination. Conversely, this study focused only on the application of one stress, which was the foliar Selenite application. This could explain the differences between the aforementioned studies and the results obtained here. In addition, it is important to consider that, although selenium is commonly considered an antioxidant at low concentrations, some studies have demonstrated that at high doses this mineral enhances oxidative stress [48,49,50]. The pro-oxidant effect of selenium was illustrated [49] by applying this mineral in excessive doses to ryegrass. Treatments with doses higher than 10 mg/kg of selenium showed a considerable increase in lipid peroxidation, despite the presence of α-tocopherols. 

In this study, the seed oils obtained from selenized plants showed the lowest OIT, presumably due to the pro-oxidant reactions promoted by the mineral, which enhanced the accumulation of lipid peroxidation products. Additionally, a decrease in total tocopherols and a higher percentage of polyunsaturated fatty acids (as described in detail in the following section) were observed likely as a response to the stress induced by selenium, but with no statistical difference compared to the control oil. Therefore, the foliar application of high doses of this mineral probably affected the plants by increasing ROS and negatively impacting OIT values.

The OIT values shown in Figure 2 (45.02, 43.67 and 39.09 min for control, 200 and 300g/Ha) were compared with commercial soybean oils, e.g., [51] reported soybean oil OIT values of 65.55 min measured by DSC at 120 °C, while, according to [52], refined, bleached, and deodorized soybean oils showed induction periods of 9.4 and 6.62 h. The differences with the results obtained in our study could be due to the method selected for the analysis and to the processes to which the oils were subjected. The oils analyzed in the present study were crude oils.

To understand the observed OIT differences, the fatty acid composition was assessed. The results are described in the following section.

### 4.4. Fatty Acid Profiles

The fatty acid composition of the three soybean oils was assessed to find out if the foliar application of selenium affected the ratio of unsaturated and saturated fatty acids and therefore OIT values since it is well known that oil oxidation is strongly associated with the amounts of polyunsaturated and monounsaturated fatty acids.

As expected, fatty acids were present in the highest amounts, particularly the polyunsaturated linoleic (35–60%) and linolenic (2–13%) and the monounsaturated oleic (20–50%) [30].

The saturated fatty acids were found in a range from 10 to 19% in soybean oils, as determined in this study. The control and selenized oils contained similar amounts of saturated fatty acids. Likewise, the polyunsaturated oleic, linoleic, and linolenic acids were present in the ranges commonly reported for soybean oil [30]. Results clearly indicate that the proposed biofortification with sodium selenite did not alter the composition of fatty acids, except for the slightly higher polyunsaturated fatty acids observed in oils obtained from seeds treated with the higher selenization.

The high concentration of polyunsaturated linoleic and linolenic acids has nutritional advantages, as well as functional disadvantages. In nutrition, the fatty acids consumption through the diet is essential since the human body lacks the necessary enzymes for their synthesis. However, from the functional point of view, a high content of polyunsaturated fatty acids makes oils more prone to oxidation. Antioxidants are added in some occasions, since, even at low levels of oxidation, off-flavors like fish or paint are generated [21]. The change in flavor generated at low levels of oxidation is known as “flavor reversion”. This phenomenon is of great concern for the soybean oil industry and is closely related to the concentration of linolenic acid and LOX activity [53,54,55].

Considering that the oils extracted from seeds selenized with 200 and 300 g Selenite/Ha contained 1.22 and 2.04% higher polyunsaturated fatty acids compared to the control, this could have influenced the observed lower OIT values (Figure 2). This is because higher contents of polyunsaturated fatty acids make oils more susceptible to oxidation [30].

### 4.5. Tocopherols and Phytosterols in Soybean Oil

Regarding tocopherols, the α moieties are regularly present in soybean oils in a range of 44 to 158 mg/kg, whereas our results, summarized in Figure 3, indicated higher concentrations, from 170.6 to 211.2 mg/kg. These concentrations are similar to values reported by [49] who studied the pro-oxidant effects of selenium over ryegrass, and by [56], who studied three different genotypes of soybean (IC210, NRC107, and JS95-60) grown in India. For γ tocopherols, the literature states their presence in a range from 926 to 1559 mg/kg; by contrast, soybean oils studied here had lower values ranging from 777.9 to 862.5 mg/kg, similar to the results obtained by [56], who found values ranging from 591.3 to 895.2 μg/g. For δ tocopherol, values are normally in the range from 254 up to 477 mg/kg [57], while this study found δ tocopherols in concentrations of 157.7 to 163.3 mg/kg; however, they were similar to those reported by [56] for soybean genotype JS95-60 (166.4 μg/g). β tocopherols were found in typical values, from 17 to 27 mg/kg, in accordance with the results described by [58] in soybeans from Brazil (6–64 mg/kg) and USA (2–29 mg/kg).

The tocopherol content can be reduced in damaged seeds and also when stored in inadequate conditions, such as high temperatures or prolonged time. The conversion of crude oil to a refined oil is also known to lower tocopherols, especially during the bleaching and deodorization steps [57,59]. In this study, the reduction in tocopherols content could have been due to unfavorable conditions during storage and the time elapsed between the extraction and analysis, but not likely to the biofortification with selenium, since there were no significant differences among treatments. Furthermore, other authors have associated the differences of tocopherols content in diverse soybean varieties to their unique genetic composition and ecological variations [58,60]. For example, the authors in [56] found similar results for γ and δ tocopherol in genotypes IC210 and JS95-60, respectively, whereas the authors in [58] described similar values to those found in our study for α and β tocopherol in soybean cultivars grown in different countries (Canada, Brazil, and USA). The similarities between the different studies suggest that the variations in tocopherols were normal and depended on diverse factors such as the soybean variety and the agro-climatic conditions at which the plants were exposed.

High tocopherol levels in soybean oil are associated with higher oxidative stability. However, higher tocopherol levels can form peroxyl, oxy, or hydroxyl radicals or singlet oxygen radicals. Thus, they become pro-oxidants and the oil can oxidize more rapidly [61]. The tocopherol content of soybeans, on the other hand, is important nutritionally because vitamin E is considered the second most relevant mechanism of defense against free radicals and oxidative stress [34]. Although tocopherols compose only approximately 1.5% of the oil extracted from soybean seeds, they are critical due to their oxidative stability. The main function of tocols is to prevent polyunsaturated fatty acid oxidation, and α tocopherol is biologically the most bioactive form of vitamin E. Additionally, it diminishes the risk of some chronic diseases such as type II diabetes and cancer. According to [62], most of the adults in the USA fail to ingest the daily requirements of vitamin E. Considering all of this, there is special interest in increasing the content of tocopherols in food. One of the strategies is to develop cultivars rich in α tocopherols, and, since soybean oil is rich in polyunsaturated fats, beneficial to human health, the selenized oils studied here represent potential new sources.

Besides tocopherols, phytosterols are other minor and relevant components of crude soybean oil together with phospholipids (collectively named lecithin). Sterols are hypocholesterolemic, that is, they significantly reduce plasma LDL levels since they have similar structure to cholesterol and therefore compete with this molecule in the specific absorption sites of the intestine [35,36,63,64]. Moreover, they possess anti-inflammatory, immunomodulatory and anticancer effects [65]. Our results, however, clearly show that the biofortification with selenium did not significantly affect the amounts and profiles of phytosterols. Total phytosterol concentration (Figure 4) has been reported previously with values up to 4050 mg/kg [63,66]. The phytosterol contents for the control and 300 g/Ha treatment are in accordance with previous reports. Although the treatments did not show statistical differences, some changes were noticed in the 200 g/Ha treatment, where campesterol and Δ5-avenasterol concentrations were slightly higher compared to the other treatment and control. However, these concentrations were within the ranges reported by [66]. The most abundant sterols in soybean are *beta*-sitosterol, campesterol, and stigmasterol. Results of this study indicated that these three sterols comprised up to 56.12, 22.66, and 17.16% respectively, of the total phytosterols in the oil. These values are within the ranges reported by [63] and consistent with previous reports for soybean oil [57,65,67].

## 5. Conclusions

The effect of selenium on the oxidative stability and composition of soybean oil was assessed after the foliar application of two different sodium selenite levels (200 and 300 g of Selenite/Ha) applied twice post anthesis. Agronomic traits and seed characterization were also evaluated. The seed chemical characteristics were not affected by the selenium fertilization, with the exception of protein content, in which the control or 0 g Se/Ha (37.6%) and 200 g Selenite/Ha (37.5%) were similar, whereas the 300 g Se/Ha treatment (35.9%) showed a slightly lower protein content, suggesting that this mineral somehow interfered with protein anabolism. Biofortification with sodium selenite represented an efficient way to increase selenium concentration in soybeans: experimental seeds contained 17.94 and 22.35 times more selenium compared to the control seed without causing adverse effects during plant development. However, selenium absorption rates (selenium absorbed compared to selenium applied in field) averaged 5%, considered low in comparison to other more effective selenization methods such as sprouting or germination (80% selenium retained by the sprouts). The use of sodium selenite reduced OIT and these values were inversely associated with the Se dose applied, suggesting that this mineral acted as pro-oxidant when applied in concentrations of 200 or 300 g/Ha. Fatty acids’ compositions were similar although selenized oils contained slightly higher amounts (between 1.22 and 2.04%) of polyunsaturated fatty acids. These results could have contributed to the OIT reduction observed because, at higher polyunsaturated fatty acids’ concentration, there is more susceptibility to oxidation. Tocopherols analysis showed a reduction in the fractions with a higher antioxidant activity (γ and δ), which could have also influenced the OIT values observed in oils extracted from selenized seeds. As for the phytosterols present in oil, no statistical differences were found. The twice-foliar application of sodium selenite in doses of 200 or 300 g Selenite/Ha post anthesis represents a viable alternative for increasing selenium concentration in soybean seeds. The best selenite level was 200 g/Ha, as there were no statistical differences in physical and chemical compositions between this treatment and control. The proposed foliar fertilization post anthesis with sodium selenite of soybeans represents an excellent alternative to produce seeds with nutraceutical potential. However, further studies are required to evaluate different doses of selenium for soybeans or other crops without enhancing lipid peroxidation.

## Figures and Tables

**Figure 1 biomolecules-09-00772-f001:**
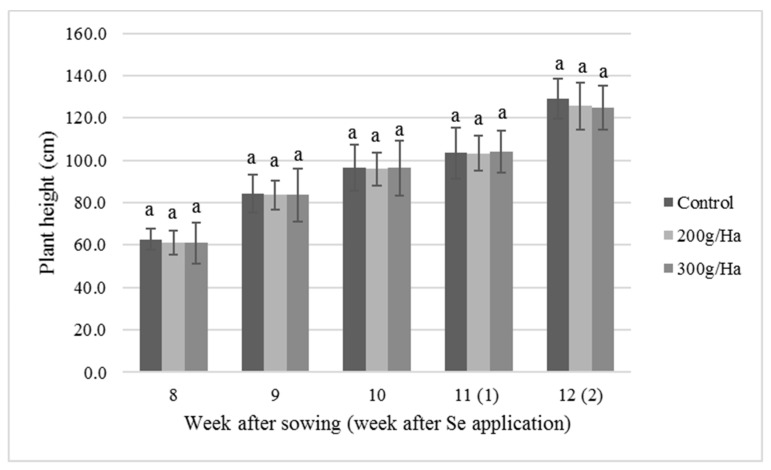
Comparison of the progressive heights of soybean plants at different weeks after sowing (in parenthesis, weeks after selenium application). subjected to different foliar selenium fertilization rates. Values are means, and bars indicate standard deviations (*n* = 24). The same letter within the week indicates no statistical difference (Tukey’s test *p* < 0.05).

**Figure 2 biomolecules-09-00772-f002:**
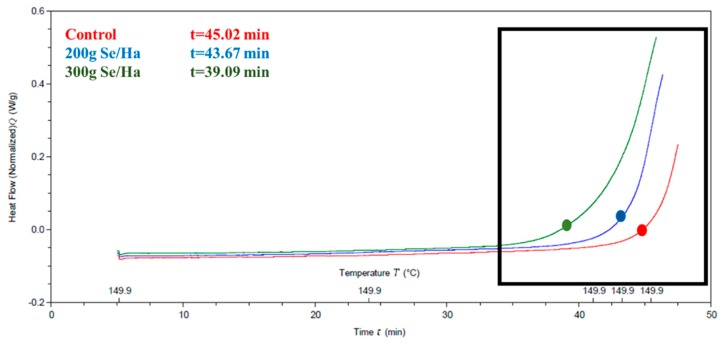
Effects of foliar application of selenium to soybean plants on the Oxidation Induction Time (OIT) of their oil.

**Figure 3 biomolecules-09-00772-f003:**
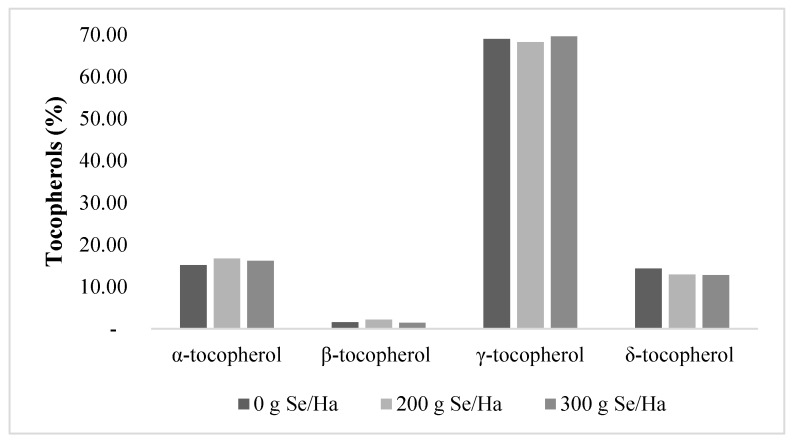
Effects of foliar application of selenium to soybean plants on the tocopherol content and profile of oils extracted from seeds.

**Figure 4 biomolecules-09-00772-f004:**
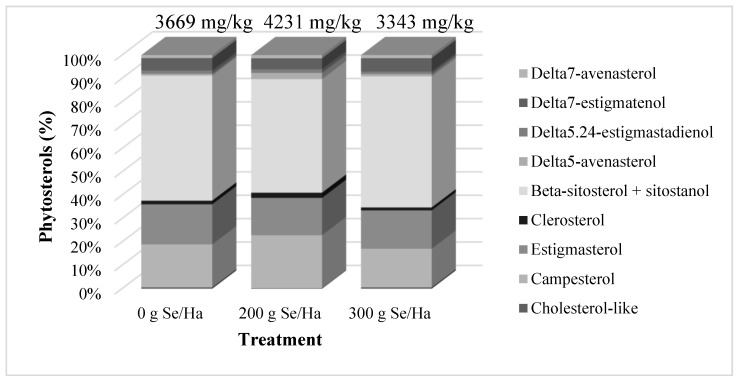
Effects of foliar application of selenium to soybean plants on the phytosterol content and profile of oils extracted from seeds.

**Table 1 biomolecules-09-00772-t001:** Effects of foliar application of selenium to soybean plants on yield, bulk and true densities and thousand weight of seeds.

Treatment	Yield,	Bulk Density,	True Density,	Thousand Seed
	kg/Ha	kg/m^3^	kg/m^3^	Weight, g
Control	1179.8 ± 80.1 ^a^	693.3 ± 5.8 ^a^	1242.1 ± 2.2 ^a^	122 ± 3.9 ^a^
200 g/Ha	1268.1 ± 167.5 ^a^	700.0 ± 10.0 ^a^	1240.5 ± 0.8 ^a^	120 ± 8.3 ^a^
300 g/Ha	1209.2 ± 206.5 ^a^	700.0 ± 10.0 ^a^	1240.0 ± 0.9 ^a^	115 ± 1.3 ^a^

Values are the average ± standard deviation (*n* = 3). Similar letters within columns indicate no statistical difference (*p* < 0.05) using Tukey’s test.

**Table 2 biomolecules-09-00772-t002:** Effects of foliar application of selenium to soybean plants on the selenium content and chemical characterization of seeds: crude fat, crude fiber, ash, protein and nitrogen free extract (non-fibrous carbohydrates).

Treatment	Crude Fat ^1^	Crude Fiber ^1^	Ash ^1^	Crude Protein ^1^	Nitrogen Free Extract ^1,2^	Selenium Content µg/g seed
Control	20.8 ± 0.1 ^a^	9.3 ± 1.4 ^a^	7.1 ± 0.0 ^a^	37.6 ± 0.5 ^a^	25.2 ± 1.6 ^a^	1.01 ± 0.14 ^a^
200 g/Ha	22.0 ± 0.1 ^a^	9.0 ± 0.5 ^a^	7.2 ± 0.1 ^a^	37.5 ± 0.2 ^a^	24.3±0.0 ^a^	18.94 ± 0.73 ^b^
300 g/Ha	22.1 ± 1.0 ^a^	8.4 ± 1.1 ^a^	7.2 ± 0.1 ^a^	35.9 ± 0.3 ^b^	26.4±0.8 ^a^	23.35 ± 0.25 ^c^

^1^ Results expressed in dry basis. ^2^ Nitrogen Free Extract (NFE) calculated as difference = 100 − (crude fat + crude fiber + ash + crude protein). Same letters within columns indicate no statistical difference (*p* < 0.05) using Tukey’s test.

**Table 3 biomolecules-09-00772-t003:** Effects of foliar application of selenium to soybean plants on density, refractive index, iodine index and saponification value of oils extracted from seeds.

Treatment	Density, g/mL	Refractive Index, RI	Iodine Index, %	Saponification Value, mg KOH/g Sample
Control	0.9029 ± 0.003 ^a^	1.4709 ± 0.003 ^a^	118.07 ^a^	192.42 ^a^
200g Se/Ha	0.9097 ± 0.005 ^a^	1.4685 ± 0.003 ^a^	118.60 ^a^	193.18 ^a^
300g Se/Ha	0.9045 ± 0.004 ^a^	1.4718 ± 0.001 ^a^	119.73 ^a^	193.39 ^a^

Same letters within columns indicate no statistical difference (*p* < 0.05) using Tukey’s test.

**Table 4 biomolecules-09-00772-t004:** Effects of foliar application of selenium to soybean plants on the fatty acid profile of oils extracted from seeds.

Fatty Acid (% w/w)	Control	200g Se/Ha	300g Se/Ha
**C12:0 Lauric**	0.27	0.14	0.12
**C14:0 Myristic**	0.30	0.17	0.18
**C16:0 Palmitic**	11.81	11.93	11.70
**C18:0 Stearic**	3.42	3.89	4.18
**C20:0 Arachidic**	0.51	0.53	0.52
**C22:0 Behenic**	0.65	0.64	0.67
**C24:0 Lignoceric**	0.31	0.33	0.32
**Saturated**	17.27	17.63	17.69
**C16:1 Palmitoleic**	0.12	0.15	0.11
**C18:1 Oleic**	33.85	32.26	31.31
**C20:1 Eicosenoic**	0.33	0.34	0.40
**Monounsaturated**	34.30	32.75	31.82
**C18:2 Linoleic**	42.70	44.17	44.43
**C18:3 Linolenic**	5.53	5.25	5.84
**Polyunsaturated**	48.23	49.42	50.27

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
