# Peer review of "Effects of Post Anthesis Foliar Application of Sodium Selenite to Soybeans (Glycine max): Lipid Composition and Oil Stability"

_biomolecules, 2019, doi:10.3390/biom9120772_

Round 1

Reviewer 1 Report

The authors present a work focused on the use of selenium salts as foliar fertilizers applied to soybean. In my opinion the authors should better explain the advantages of the use of selenium and the quantities proposed since the only significant variation observed is the concentration of this element in the seeds and oil of soybeans. Furthermore, the manuscript should be reviewed in depth by a native English speaker. To achieve the quality standards required by Biomolecules, the manuscript in this version would need a thorough revision by following the suggestions above.

Author Response

Thanks a lot. Please see the attachment. 

Reviewer 2 Report

The connection results with discussion will be beneficial to the article in my opinion. Peroxide value of oils from control and fertilized seeds have to be determined and discussed. The methodology about Oxidation Induction Time (OIT) is incomplete. Which gas was used? What was the flow rate of gas? What kind of pan was used in experiments and from what material pans were made of? The rate of 20°C per minute was too high in my opinion. Figure 2 has an additional axle with temperature, what does it mean? The discussion about OIT requires connection to discussion about fatty acid profiles. 

Author Response

Thanks a lot for your comments. Please see the attachment. 

Round 2

Reviewer 1 Report

The authors have diligently followed the suggestions of the reviewers improving the general quality of the manuscript. In the current version the manuscript is worthy of being published on Biomolecules

Reviewer 2 Report

Effects of Post Anthesis Foliar Application of Sodium Selenite to Soybeans (Glycine max): Lipid Composition and Oil Stability

María-José Escalante-Valdez, Daniela Guardado-Félix, Sergio O. Serna-Saldívar, Daniel Barrera-Arellano, Cristina Chuck-Hernández

The manuscript is written in a correct English language and the results are clearly discussed. All of my suggestions have been included by the Authors. Discussion is very interesting. I approve of all corrections.